# Long-Term Adverse Effects of Mild COVID-19 Disease on Arterial Stiffness, and Systemic and Central Hemodynamics: A Pre-Post Study

**DOI:** 10.3390/jcm12062123

**Published:** 2023-03-08

**Authors:** Mario Podrug, Pjero Koren, Edita Dražić Maras, Josip Podrug, Viktor Čulić, Maria Perissiou, Rosa Maria Bruno, Ivana Mudnić, Mladen Boban, Ana Jerončić

**Affiliations:** 1Laboratory of Vascular Aging, University of Split School of Medicine, 21000 Split, Croatia; 2University Department of Health Studies, University of Split, 21000 Split, Croatia; 3University of Split School of Medicine, 21000 Split, Croatia; 4Infectious Diseases Department, University Hospital of Split, 21000 Split, Croatia; 5Otorhinolaryngology Department, University Hospital of Split, 21000 Split, Croatia; 6Department of Cardiology and Angiology, University Hospital Centre Split, 21000 Split, Croatia; 7Physical Activity, Health and Rehabilitation Research Group, School of Sport, Health and Exercise Science, Faculty of Science and Health, University of Portsmouth, Portsmouth PO1 2UP, UK; 8Université Paris Cité, INSERM, PARCC, 75015 Paris, France; 9Clinical Pharmacology Unit, AP-HP, Hôpital européen Georges Pompidou, 75015 Paris, France; 10Department of Basic and Clinical Pharmacology, University of Split School of Medicine, 21000 Split, Croatia; 11Department of Research in Biomedicine and Health, University of Split School of Medicine, 21000 Split, Croatia

**Keywords:** arterial stiffness, central hemodynamics, COVID-19, vascular remodeling, long COVID-19 syndrome, autoimmune response

## Abstract

COVID-19-associated vascular disease complications are primarily associated with endothelial dysfunction; however, the consequences of disease on vascular structure and function, particularly in the long term (>7 weeks post-infection), remain unexplored. Individual pre- and post-infection changes in arterial stiffness as well as central and systemic hemodynamic parameters were measured in patients diagnosed with mild COVID-19. As part of in-laboratory observational studies, baseline measurements were taken up to two years before, whereas the post-infection measurements were made 2–3 months after the onset of COVID-19. We used the same measurement protocol throughout the study as well as linear and mixed-effects regression models to analyze the data. Patients (N = 32) were predominantly healthy and young (mean age ± SD: 36.6 ± 12.6). We found that various parameters of arterial stiffness and central hemodynamics—cfPWV, AIx@HR75, and cDBP as well as DBP and MAP—responded to a mild COVID-19 disease. The magnitude of these responses was dependent on the time since the onset of COVID-19 as well as age (p_regression_models_ ≤ 0.013). In fact, mixed-effects models predicted a clinically significant progression of vascular impairment within the period of 2–3 months following infection (change in cfPWV by +1.4 m/s, +15% in AIx@HR75, approximately +8 mmHg in DBP, cDBP, and MAP). The results point toward the existence of a widespread and long-lasting pathological process in the vasculature following mild COVID-19 disease, with heterogeneous individual responses, some of which may be triggered by an autoimmune response to COVID-19.

## 1. Introduction

In December 2019, the first official case of coronavirus disease (COVID-19) was detected in the Chinese city of Wuhan. Since its first breakout, the virus has swiftly spread over the world, and the World Health Organization (WHO) declared a pandemic in March 2020. At the time of writing, there had been 660,378,145 confirmed cases of COVID-19, with 6,691,495 fatalities globally (1). While COVID-19 was initially thought of as an acute respiratory illness, it is now recognized as a complex multisystemic disease with extensive and deleterious cardiovascular involvement [1,2]. In addition to direct consequences and complications due to acute COVID-19 infection, a recent study showed that 12 months after the onset of COVID-19 infection, up to 25% of patients who were otherwise healthy and free of underlying diseases exhibited the long COVID-19 syndrome [3]. 

Subclinical myocardial and vascular dysfunction have been linked to worse outcomes and an increased risk of death in patients with COVID-19 disease [4]. Even in patients with mild COVID-19 disease severity, the infection has been linked to impaired subclinical markers of cardiovascular and endothelial function [5]. It is presumed that COVID-19-associated vascular disease complications may be precipitated by direct endothelium damage [6] or immune-mediated vascular damage [7,8]. However, it is unknown to what extent structural alterations of the vascular wall occur in addition to endothelial damage. Even fewer data exist regarding the long-term effects of COVID-19 infection on vascular structure and function. Current fragmented evidence suggests that COVID-19 disease reduces systemic vascular function and increases arterial stiffness [9,10].

Arterial stiffness is a vascular aging phenomenon that refers to a loss of arterial compliance or changes in artery wall characteristics [11]. Arterial stiffness worsens with age and exposure to risk factors that hasten the stiffening process [12,13]. Various measures of arterial stiffness and central hemodynamics can reveal a decline in arterial elasticity brought on by structural wall changes in the arterial system. The most validated and direct measure of arterial stiffness is the carotid–femoral pulse wave velocity (cfPWV) (Townsend, Wilkinson et al., 2015). In addition, augmentation indices are indirect measures of arterial stiffness which are believed to capture the negative impact of systolic wave reflection on cardiac workload [14]. Finally, the central blood pressures refer to the pressure in the ascending aorta. These are the pressures that the target organs are subjected to, and they are lower than brachial cuff pressures due to arterial pressure amplification [15].

We hypothesized that even mild cases of COVID-19 disease could have long-term detrimental effects on arterial structure and function. To investigate this, we examined individual pre- and post-infection changes in arterial stiffness as well as systemic and central hemodynamic parameters in patients diagnosed with mild COVID-19. Baseline measurements were taken up to two years before a participant became infected, and post-infection measurements were taken two to three months after the onset of the disease.

## 2. Materials and Methods

### 2.1. Study Design

This is a pre–post study design in which measures of arterial stiffness and central hemodynamic were recorded in a group of participants before and after the COVID-19 infection. 

All the recordings were made between October 2019 and April 2022 in the Laboratory for Vascular Aging at the University of Split School of Medicine. 

To assess arterial stiffness and central hemodynamic parameters prior to COVID-19 infection, we utilized the stored recordings of enrolled participants from in-laboratory observational studies that applied the same measurement protocol as was used for the post-COVID-19 measurements. The post-COVID-19 measurement was performed between 8 and 12 weeks after the COVID-19 infection had ended, as evidenced by the absence of symptoms. This timeframe corresponded to 50 ± 2 to 90 ± 2 days after the onset of the first symptoms. The maximum amount of time between pre- and post-COVID measurements was set at 24 months. 

### 2.2. Participants

The participants who had their arterial stiffness and central hemodynamic outcomes measured in our laboratory prior to infection with COVID-19 and who were afterwards infected with the virus were considered eligible for inclusion in the study. For all of those invited to the study, COVID-19 diagnosis was made by real-time Polymerase Chain Reaction test. During their first visit to the laboratory (pre-COVID measurements), all the participants underwent a medical history, and those with arrhythmias, cerebrovascular sickness, pregnancy, surgery amputation, oncology disease, psychiatric disease, infections throughout the trial duration, medical nonadherence, those that were unable to provide fully informed written consent or had any other serious medical condition that may affect data interpretation were excluded from the study.

While this was not originally our inclusion criteria, all of the participants reported mild severity of COVID-19.

In total, we invited 36 adults to participate in our study, with 32 (89%) agreeing to take part. All participants provided written informed consent to participate in the study, which conformed to the Declaration of Helsinki. 

### 2.3. Study Procedures

Before undergoing testing, participants filled out a health history questionnaire, which inquired about personal and family medical history as well as medication use. They arrived for testing in a fasted state having abstained from food, caffeine, or smoking for at least 3 h and from exercise, alcohol and smoking for 24 h before testing. Those taking vasoactive medications (3 or 9%) maintained the same dosage throughout the duration of the study.

All study procedures were carried out in a quiet, temperature-neutral environment with the temperature range of 21–23 °C after participants had lain supine for 10 min.

To avoid possible confounding, each participant was recorded at the same time of day and with the same device during both visits.

### 2.4. Study Measurements

The arterial stiffness and central hemodynamics measurements were taken in accordance with the American Heart Association’s recommendations for improving and standardizing vascular research on arterial stiffness (Townsend, Wilkinson et al., 2015).

Office blood pressure (BP) was measured during each visit using the validated oscillometric sphygmomanometer (Welch Allyn Connex ProBP 3400 digital blood pressure monitor with SureBP technology). The BP measurements were taken in a supine position after 5 min of resting and prior to PWV measurements. The participants did not change body posture between the two measurements. 

Carotid–femoral pulse wave velocity (cfPWV); central blood pressures including: central systolic (cSBP) and diastolic (cDBP) blood pressures and pulse pressure (cPP); pulse pressure amplification; augmentation pressure (AP); augmentation indices: AIx calculated as AP/PP, AIx©75—AIx calculated as AP/PP and normalized to the heart rate of 75 beats per minute (bpm), and AIx index calculated as the ratio of late to early systolic pressure P2/P1; and heart rate (HR) were measured by either applanation tonometry using the Sphygmocor CvMS device (Atcor Medical, Sydney, Australia) or by the hybrid applanation tonometer—oscillometric device SphygmoCor Xcel (Atcor Medical, Sydney, Australia), as described previously [16,17]. While the validation studies comparing two devices indicated that they were comparable in terms of assessment of carotid–femoral pulse wave velocity (cfPWV) and augmentation index (AIx) [18,19], each participant was recorded using only one device to ensure that intra-individual changes are not affected by the type of device used.

A single operator (M. P.) carried out all of the measurements. For cfPWV measurements, recordings were performed on the right carotid and the right femoral artery. Central BPs and other parameters derived from the pulse wave analysis (PWA) were estimated after calibration of the pulse waveform recorded at the radial artery to mean and diastolic brachial pressures. We used the subtracted distance method to calculate the wave travel distance. The method was chosen over the direct method as per recommendation by the latest guideline [20]. 

### 2.5. Sample Size Considerations

Thirty-two participants are sufficient to detect moderate to strong effects on parameter changes using a simple linear regression model and strong effects when two-predictor linear regression model is used. Namely, using a two-predictor multiple linear regression model with α = 0.05, f^2^ = 0.35, and power of 80%, a sample size of 31 is required to detect a strong association between pre–post changes in vascular function/structure and potential predictors. For estimations based on a simple linear regression model, the sample size of 32 was sufficient to detect moderate to strong associations under assumptions of α = 0.05, f^2^ = 0.27, and power of 80%. 

### 2.6. Data Analysis

To describe the distribution of a quantitative variable, we used mean and standard deviation (SD) or median and interquartile range (IQR), depending on the shape of the distribution. To decide if a distribution is asymmetrical, we used skewness and kurtosis tests for normality. The distribution of a qualitative variable was described with absolute and relative frequencies.

We utilized one-sample tests to determine whether a pre–post change in a parameter was statistically significant: either the parametric one-sample *t*-test or its non-parametric counterpart, the sign rank test, depending on the symmetry of the variable’s distribution.

There were two sets of the regression models developed. We employed simple or multiple linear regression (MLR) models to identify predictors of the change from baseline (pre-COVID) for different arterial stiffness and hemodynamic parameters. These models were preferred as they use individual pre–post changes as the dependent variable. In addition, we built mixed-effects regression models to identify predictors affecting the values of a modeled parameter. Due to the fact that mixed-effects regression models employ repeated measurements of a parameter, these models had greater analytical power than simple or MLR models.

The model building was performed in two steps. In the first step, potential predictors—including age, sex, the amount of time that passed since the start of COVID infection, the amount of time that passed between the pre- and post-COVID-19 measurements, pre-COVID baseline values of a modeled parameter, and the type of device used to estimate its values—were used as single predictors in a simple linear regression or a mixed-effects model. For the final model, only those predictors that were significant at the 0.1 level or higher were considered (*p* ≤ 0.01). Requirements for inclusion in the final model were significance at the 0.05 level or an increase in adjusted R^2^ of at least 2% and a *p*-value of less than 0.2.

The above-mentioned potential predictors that were initially evaluated were selected to estimate the dependence of parameter values on the time that passed since the COVID infection and account for potential confounding variables. As an example, even though we did not anticipate any significant changes in vascular function over a 24-month period in the predominantly young participants (Table 1), we included the time between the first and second measurements as a potential predictor to control for its confounding effect. 

We interpreted the strength of association between a predictor and a modeled parameter by applying the Cohen’s effect size magnitudes for R^2^ (small from 0.02 to <0.13, medium from 0.13 to <0.26, large from ≥0.26) to the adjusted R^2^ of a single predictor model. [21] 

As this is an exploratory study, no control for multiple testing was performed. The analysis was performed in STATA (version 17.0, Stata Corp. LP, College Station, TX, USA). We applied the significance level of *p* = 0.05. 

## 3. Results

In this study, 32 participants were recruited; each participant visited the laboratory twice; and all of their data were collected. Table 1 shows their demographic and clinical characteristics at baseline prior to the COVID-19 infection. The participants were predominantly young (≤40 years) and healthy with only 9% (n = 3) of the cohort being hypertensive. None of the participants had dyslipidemia, and only two had diabetes (6%, one person was also hypertensive). The majority of the cohort was overweight or obese (69%) and did not smoke (78%). 

The average time since the onset of COVID-19 infection in our sample was mean ± SD: 73 ± 10 days, with this time ranging from 51 to 92 days. The median time that elapsed between two measurements was 327.5 days (IQR, 129 to 458), with the range between 74 and 730 days. The majority of participants, 23 or 72%, were recorded with the SphygmoCor XCEL device. 

In terms of the severity of COVID-19 infections, none of our participants have developed any of the cardiovascular, pulmonary, thromboembolic, or other COVID-19-associated complications, and there were no hospitalizations. Participants were evenly distributed according to the year they became infected (chi-square test, *p* = 0.084). There was no significant pre–post change in weight: median change 0 kg, 95% CI from −0.3 to 0.5.

When we looked to see if the mean individual pre–post changes were significantly different from 0, we found no significant pre–post change in any of the arterial stiffness or hemodynamic parameters tested (*p* ≥ 0.122). We did, however, see an average increase of 0.19 m/s in carotid–femoral pulse wave velocity (cfPWV) from pre-infection values but at the significance level of 0.1 (*p* = 0.052). Table 2 shows the distribution of vascular parameters at baseline and after the infection.

Further analysis, however, revealed a widespread and complex pattern of confounders that affect the pre–post infection changes, and parameter values, in the majority of the assessed arterial stiffness and hemodynamic parameters (Table 3 and Table 4). The size of these changes, as well as the direction in which they went, were dependent not only on the length of recovery time that had passed since the onset of the COVID-19 infection, confirming the existence of a response to the COVID-19 infection, but also, and more commonly, on cardiovascular health status at baseline (ascertained by age of an individual at baseline or the value of a parameter at baseline), as well as the amount of time that had passed between the two measurements. In accordance with the Cohen’s interpretation of R^2^, the majority of identified associations, regardless or the type of model, were moderate to strong [21]. 

### 3.1. Arterial Stiffness—cfPWV

Regarding the cfPWV response to COVID-19 infection, defined as the pre–post change in this parameter, we found that post-infection values increased by 0.19 m/s (95% CI −0.04 to 0.41) but only at the significance level of 0.1 (*p* = 0.052). We also found no evidence that age, time since the onset of COVID 19, time between measurements, or cfPWV baseline values influence individual cfPWV responses. Individual pre–post changes were also significant at the 0.1 level according to the mixed-effects model (Table 4). However, age and time were moderately and positively associated with the cfPWV change since the onset of COVID infection at a group level. This model, which explains 32% of the intra-individual variability and 28% of variation at the group level, predicts an increase of 1.14 m/s in the average cfPWV value as a result of variation in the time since the onset of COVID-19 infection (51–92 days). The relationship between cfPWV and two predictors is shown in Figure 1. Although the age dependence is to be expected, we included it for comparison purposes. The change in cfPWV was not determined by the pre–post change in HR or the baseline HR value, nor was it affected by the change in BMI or the baseline BMI value (*p* ≥ 0.308).

### 3.2. Arterial Stiffness—Augmentation Indices

As previously stated, no significant pre–post changes in augmentation indices were observed following the COVID-19 infection (*p* ≥ 0.244). However, we discovered that the pre–post changes increased with age in AP and all of the AIx indices: Aix AP/PP, Aix P2/P1, and AIx@HR75 (Table 3). Except for the pre–post change in Aix P2/P1, which was moderately associated with age, this dependency was generally weak (Table 3). 

Aside from age, which was found to be a common predictor of AIx pre–post changes, we discovered additional time-related predictors of these changes in Aix P2/P1 and AIx@HR75 indices. 

Time since the onset of COVID-19 infection was a moderate and positive predictor of pre–post changes in the AIx@HR75, accounting for 20% of their variance (Table 3, Figure 2). Within a range of 51 to 92 days after the onset of COVID infection, the AIx@HR75 pre–post change was predicted to move from −5% to +10%. 

### 3.3. Peripheral and Central Hemodynamics

We found no significant changes from baseline in any peripheral or central hemodynamics parameters (*p* ≥ 0.122). 

Pre–post changes from baseline in SBP, cSBP, PP, and cPP were negatively dependent on their baseline values (weak—cSBP and cPP, moderate—SBP, strong—PP; Table 3). For example, given the range of baseline values for the PP parameter, the predicted post-COVID increase in PP ranges from +4 to -11 mmHg. It should also be noted that we also found a significant association of pre–post changes in PP with the time from onset of COVID-19, but at a 0.1 level of significance (*p* = 0.098). The variable was not included in the final model of pre–post PP changes because it did not meet our protocol’s inclusion requirements.

Using the mixed-effects models, we were able to identify that age, and to a lesser extent the time since acquiring COVID-19, were positively associated with parameter values for DBP, cDBP, MAP, and cfPWV (Table 4). According to these models, the estimated change in average pressure values caused by variations in the amount of time that has passed since the beginning of the COVID-19 infection is as follows: DBP is envisaged to increase by 8.1 mmHg, cDBP by 7.6 mmHg, and MAP by 7.6 mmHg. As for individual pre–post changes, we were unable to identify significant mean changes nor the associations of these changes with any of the predictors listed above, nor were we able to find significant individual pre–post changes within mixed-effects models. Such a finding suggests that individual changes are likely heterogeneous and that that the effect that we identified at the group level is probably an average effect. 

### 3.4. The Age Dependence of Pre-Post Changes in Investigated Parameters

We examined the pattern of pre–post changes in age dependence to see if there is a possibility that age modifies responses to COVID-19 infection. The scatter plots in Figure 3a–c depict a distinct pattern for those aged under and over 40. We demonstrated that the change in AIx P2/P1 from baseline for those over 40 years old is significantly greater than 0 (median 6%, 95% CI 0.7–24%, *p* = 0.005), whereas no significant change was observed for those under 40 years old (*p* = 0.976) (Figure 3a). 

## 4. Discussion

This is the first study to compare pre- and post-COVID-19 infection levels across a wide range of arterial stiffness and hemodynamics parameters in the same group of participants. We discovered that the responses of the vascular system to a mild COVID-19 disease, defined here as systematic, individual pre–post differences in investigated parameters, are not simple in the sense that COVID-19 on average either increases or decreases a parameter in infected patients by a comparable amount of measurement units. In fact, except for a non-significant trend for cfPWV, we were unable to detect any parameter with a mean pre–post COVID-19 change that differed from 0. 

Instead, responses to COVID-19 infection are dynamic and depend on the time since the onset of COVID-19 infection. We identified such time-dependent responses in the arterial stiffness parameters—cfPWV and AIx@HR75, the central hemodynamic parameter—cDBP, and the systemic hemodynamics parameters—DBP and MAP; and we showed that their values increased with the length of time that passed from the onset of COVID-19 infection, independent of age or other confounders. In addition, the vascular impairment predicted by our models for observation period of two to three months post-infection is clinically significant as shown by an increase in cfPWV of +1.4 m/s, +15% in AIx@HR75, +8 mmHg in DBP, and +7.6 mmHg in cDBP and MAP.

The finding that the longer the period from COVID-19 infection the worse the vascular impairment was surprising, as we expected inflammation burden associated with COVID-19 to decrease with time. While we can only speculate on what causes this phenomenon, emerging evidence suggests that it stems from a failure to resolve autoantibodies observed during the acute phase of disease [22,23,24], or alternatively, that generating de novo pathogenic autoimmune responses post-recovery contributes to long COVID with evidence of residual inflammatory cytokines [25,26,27]. Hence, what we observed at the group level, 2–3 months after infection, may be related to inflammation-induced arterial stiffening in some individuals [28], which is caused by inflammation from an autoimmune response or chronic inflammation that precedes one [29]. Furthermore, the heterogeneous responses observed in our study could be explained by the fact that inflammation in post-recovery was not triggered in all patients. Indeed, a recent study found that the circulating levels of anti-/extractable nuclear autoantibodies (ANA/ENAs) were higher at 3 months post-recovery in patients who had COVID-19 and were free from autoimmune diseases at the time of infection compared to healthy and non-COVID infection groups. High circulating ANA/ENA titers, which correlated with long COVID symptoms, were maintained up to 6 months after recovery but significantly reduced by 12 months. Even after 12 months, several pathogenic ANA/ENAs were still detectable in up to 30% of COVID survivors. Furthermore, a retrospective study of 4 million participants found an increased risk of autoimmune diseases in patients with COVID-19 with an adjusted hazard ratio for different autoimmune diseases ranging from 1.78 to 3.21 [30].

All the time-dependent responses to COVID-19 disease were also affected by age in a way that each additional year at baseline added to vascular impairment post-infection. The effect of age was not the result of the time gap between pre- and post-COVID-19 measurements, as this confounding variable was controlled for in our analyses. In addition, we could not assign the effect of age only to the increased variability of investigated parameter with age [16], because in that case, pre–post differences would go in both directions—positive and negative, and we would not be able to find an increasingly positive relationship with age. Age, however, may modulate the response to a mild COVID-19 disease in arterial stiffness and central hemodynamics parameters in different age groups. Previous studies have suggested an age modulation of vascular responses to various triggers, including an infection [31,32], and the association of age with autoimmune inflammation [33]. While our results suggest the role of age as a modifiable factor in the response to mild COVID-19 disease, such a role should be further examined in studies with a larger sample size. 

We detected the responses to COVID-19 disease in a variety of arterial stiffness measures and measures of its hemodynamic consequences, including: the direct (cfPWV) and indirect (augmentation index) measures of arterial stiffness as well as central (cDBP) hemodynamic parameter. Each of these three parameters represents a distinct aspect of the atherosclerotic process, which involves morphological and/or functional alterations to the vessel wall [34]. Therefore, the simultaneous detection of responses to COVID-19 disease in various vascular structure and function parameters supports the existence of a widespread and long-term pathological process in the vasculature following infection [35].

So far, only a handful of studies investigated the effect of COVID-19 infection on arterial stiffness and central hemodynamics. Most of them were case control studies with small sample sizes (10–22 per arm) comparing patients recovering from COVID-19 with controls [4,9,10]. Despite the possibly limited power of these studies, their results support our conclusions regarding the existence of vascular impairment after COVID-19.

The fact that cfPWV is increased in participants after the COVID-19 infection when compared to controls was found in several studies performed on: young healthy patients and their controls 3–4 weeks after the onset of COVID-19 (increase of 0.7 m/s) [9], acutely ill elderly patients (increase of 3.3 m/s) [4], as well as middle-aged patients that were compared to controls at 4 months (increase of 2.05 m/s) [36] and 12 months (increase of 1.15 m/s) after the COVID-19 onset [37]. 

Aix, like cfPWV, has been found to be higher in COVID-19-infected participants compared to controls. A 10% increase in the augmentation indices AIX AP/PP and AIx@HR75 has been reported in those infected with COVID19 when comparing 15 young adults 3–4 weeks after a positive COVID-19 test to healthy young controls. [10].

Finally, in terms of cSBP, at the 4- and 12-month follow-ups, COVID-19 patients have had a persistent increase of 10 mmHg in cSBP compared to controls [36,37]. In addition, Akpek et al. [38] reported an increase in systemic hemodynamics parameters during short-term follow-up in patients diagnosed with COVID-19.

Only one case control study did not find significant differences in arterial stiffness parameters—PWV and AIx75—at 4 weeks post-infection when young adults who were infected with COVID-19 were compared with their controls [39]. In addition, two small longitudinal studies reported results contradicting our findings. In the first study that followed 14 young participants from the first to sixth month post-infection, the authors reported a decrease in cfPWV (decrease by 0.82 m/s), SBP (by 11 mmHg), MAP (by 11 mmHg) with time; and no change in time was found for AIx@HR75 [40]. The second study followed 10 young adults for 6 months after the COVID-19 infection and found that SBP and DBP decreased throughout the study: with SBP decreasing by 15 mmHg and DBP decreasing by 10 mmHg [41]. Given that both of these longitudinal studies reported participant attrition on very small sample sizes, used inappropriate statistics (mean and standard deviation) to describe the distribution of limited data, and removed outliers from a small sample size [40], the reported results could be the result of methodological issues. On the other side, the lack of a uniform individual response to COVID-19 in the investigated parameters of vascular structure and function, which may be the consequence of age modulation (and possibly modulation by other factors), may have caused such results. 

Our study had some limitations, the most significant of which was that the sample size only allowed us to detect moderate to strong associations. This means that even though we may have confidence in the significant associations observed in our study, we may have overlooked a relationship between the time since the onset of COVID-19 and several other parameters. For example, pre–post changes in PP were significantly associated with the time from COVID at the lower significance level of 0.1. As our sample size was limited by the number of recent pre-COVID recordings in our laboratory, we were not able to expand the sample size further. 

Another potential drawback is the possibility that age and perhaps other factors relate to moderate responses to COVID-19 and that certain patient subgroups respond differently to COVID-19 than it was predicted in the overall model for that parameter. The fact that our models did not detect that individual pre–post changes in the tested parameters are significantly different from 0 but were able to detect in the mixed-effects models that parameter values depend on time from COVID-19, at the level of the entire sample, suggests that heterogeneous responses are possible. If this was the case, given that our results suggested an average response to COVID-19 disease, further analyses should be performed in studies with larger sample sizes.

Finally, because MAP in our study was estimated rather than directly measured, its estimation may be unreliable for some individuals [42], which could affect the accuracy of parameters derived from pulse wave analysis. However, since we were able to identify general patterns of change and interdependence, we do not expect this had a significant impact on the results.

The findings of this study demonstrated that there is a widespread and long-lasting pathological process in the vasculature following the mild COVID-19 infection which keeps deteriorating during 2–3 months post-infection. In light of the recent finding that up to 25% of otherwise healthy and disease-free patients exhibited the long COVID-19 syndrome 12 months after the onset of COVID-19 infection [3], and in light of the fact that vascular impairment increases the risk of future cardiovascular events, it is crucial that future studies explore these changes with larger sample sizes and with more synchronous population regarding the onset of COVID-19.

## 5. Conclusions

We found that various parameters of arterial stiffness and central hemodynamics respond simultaneously to the mild COVID-19 disease in predominantly healthy individuals. While we were unable to demonstrate this effect on all of the parameters tested, the worsening of values of those found to be responsive (cfPWV, AIx@HR75, cDBP, DBP, and MAP) points toward the existence of a widespread and long-lasting pathological process in the vasculature following the infection. 

The detected responses to COVID-19 disease are not straightforward but rather deteriorate with the time since the onset of COVID-19 infection and age. 

Within the period of 2–3 months following infection, our models demonstrated a clinically significant progression of vascular impairment.

Finally, we discovered that individual responses to COVID-19 are likely heterogeneous and possibly moderated by age.

Emerging evidence suggests that post-recovery autoimmune response to COVID-19 may be the cause of this phenomenon, although we can only speculate on its origin.

## Figures and Tables

**Figure 1 jcm-12-02123-f001:**
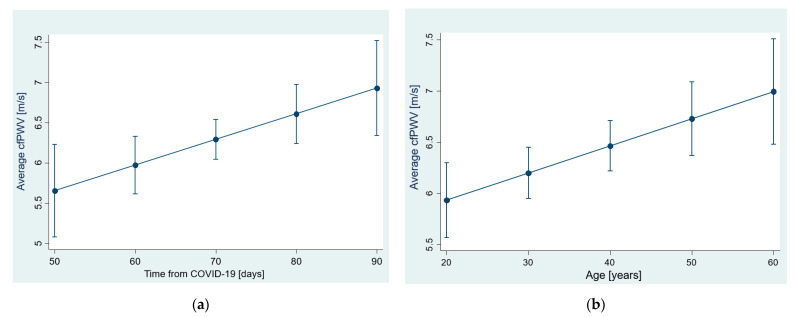
The relationship between the average cfPWV values in the sample and: (**a**) the time since COVID infection; (**b**) age, as estimated with the mixed-effects model (R^2^ = 27% at the level of sample, R^2^ = 31% for individual changes). Shown are predictive margins of cfPWV values with 95% CIs.

**Figure 2 jcm-12-02123-f002:**
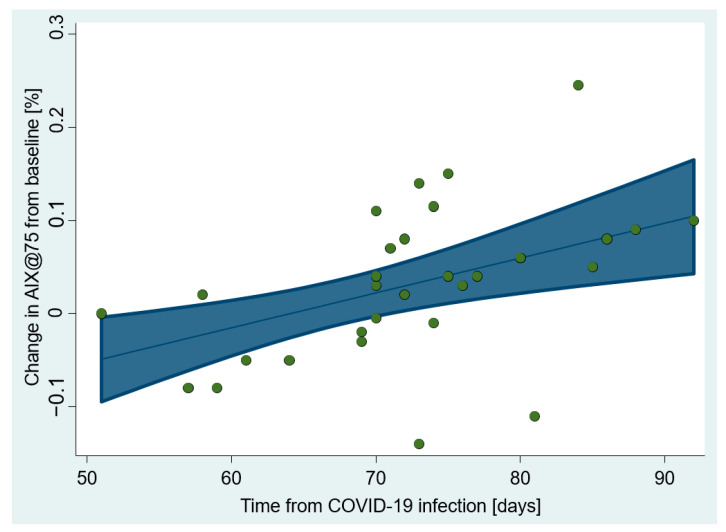
The change in AIx normalized to the heart rate of 75 bpm (AIx@HR75) from the baseline depends on the time that passed from acquiring COVID-19 infection. Shown are predictive margins with 95% CI estimated by the LR model (R^2^ = 26%) and the scatter plot of observation.

**Figure 3 jcm-12-02123-f003:**
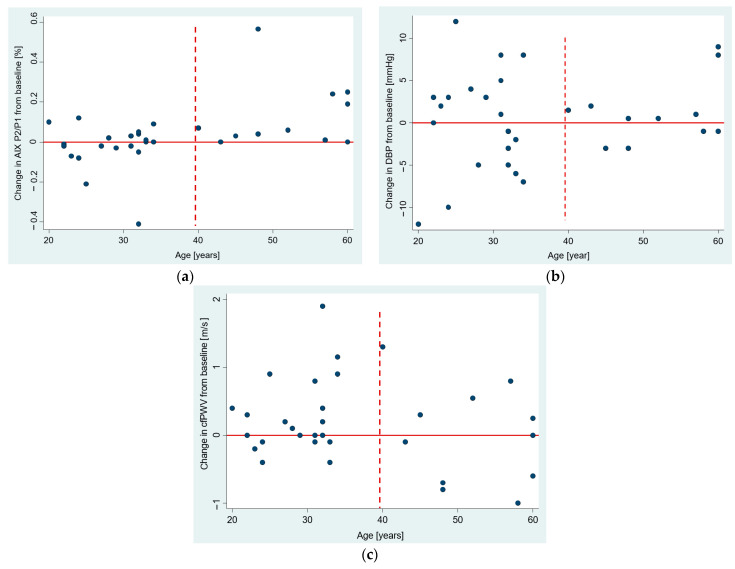
Scatter plots showing the relationship between age and pre–post change in: (**a**) augmentation index AIx P2/P1, (**b**) diastolic blood pressure, and (**c**) cfPWV. The horizontal reference line set at zero pre–post change, and the vertical dashed line set at 40 years that separates parts of a plot with apparently different dispersion patterns.

**Table 1 jcm-12-02123-t001:** Demographic and clinical characteristics of participants, N = 32.

Characteristic	Statistics
Sex, N (%) Male Female	18 (56%) 14 (44%)
Age (years), mean ± SD	36.6 ± 12.6
BMI, median (IQR)	28 (24.5 to 31.4)
Hypertension, N (%)	3 (9%)
Diabetes, N (%)	2 (6%)
Dyslipidemia, N(%)	0 (0%)
Familial history of CV disease, N (%)	7 (22%)
Smoking, N (%) No Yes Ex-smoker	17 (53%) 7 (22%) 8 (25%)
Smoking [cigarettes per day], median (range) *	5–10 cigarettes (1–10)

BMI—body mass index; * Calculated on N = 7 smokers.

**Table 2 jcm-12-02123-t002:** Distribution of arterial stiffness, and central and systemic hemodynamic parameters at baseline (pre-infection) and 2–3 months after the onset of COVID-19 disease (post-infection), N = 32.

Parameter	Pre-Infection	Post-Infection
Systemic Hemodynamics		
SBP (mmHg), mean ± SD	120 ± 9	119 ± 9
DBP (mmHg), mean ± SD	70 ± 8	71 ± 9
MAP (mmHg), mean ± SD	86 ± 8	85 ± 10
PP (mmHg), median (IQR)	47 (43, 54)	47 (43, 51)
HR (bpm), mean ± SD	65 ± 10	64 ± 7
Central Hemodynamics		
cSBP (mmHg), mean ± SD	107 ± 7	107 ± 9
cDBP (mmHg), mean ± SD	71 ± 8	72 ± 9
cPP (mmHg), mean ± SD	36 ± 6	35 ± 6
Carotid–Femoral Pulse Wave Velocity		
cfPWV (m/s), mean ± SD	6.3 ± 0.7	6.5 ± 1.0
Pulse Wave Analysis		
Aortic Augmentation (mmHg), mean ± SD	7 ± 5	7 ± 6
Aortic AIx, P2/P1 (%), mean ± SD	19% ± 13%	20% ± 16%
Aortic AIx, AP/PP (%), mean ± SD	123% ± 13%	126% ± 19%
Aortic AIx@HR75, P2/P1 (%), mean ± SD	15% ± 14%	15% ± 17%

AIx, augmentation index; AIx@HR75, augmentation index corrected for HR; cDBP, central diastolic blood pressure; cPP—central pulse pressure; cSBP, central systolic blood pressure; BP, blood pressure; cfPWV, carotid–femoral pulse wave velocity; DBP, diastolic blood pressure; HR, heart rate; IQR—interquartile range; MAP—mean arterial pressure; P1, first systolic peak; P2, second systolic peak; PP—pulse pressure; SBP, diastolic blood pressure; SD—standard deviation.

**Table 3 jcm-12-02123-t003:** The predictors of the pre–post COVID-19 changes in systemic and central hemodynamic parameters and arterial stiffness parameters, as estimated by the linear regression models.

Pre–Post Change in:	Predictor	B (95% CI)	*p*-Value	Adjusted Simple Model R^2^	Adjusted Final Model R^2^
Systemic and Central Hemodynamics					
SBP (mmHg)	Baseline value *	–0.46 (–0.68 to –0.24)	<0.001	21 §	21 §
DBP (mmHg)	no significant model				
PP (mmHg)	Baseline value	–0.35 (–0.68 to –0.02)	0.041	26 §§	30 §§
	Device, XCEL vs. CvMs	4.16 (–0.02 to 8.34)	0.051 †	12	
MAP (mmHg)	no significant model				
cSBP (mmHg)	Baseline value	–0.24 (–0.44 to –0.03)	0.026	3	3
cDBP (mmHg)	no significant model				
cPP (mmHg)	Baseline value	–0.36 (–0.71 to –0.18)	0.040	12	12
Carotid–Femoral Pulse Wave Velocity					
cfPWV (m/s)	no significant model				
Pulse Wave Analysis					
Aortic AP (mmHg)	Age	0.11 (0.02–0.21)	0.023	7	7
Aortic AIx, AP/PP (%)	Age	0.003 (0.0008–0.006)	0.013	10	10
Aortic AIx, P2/P1 (%)	Age	0.005 (0.002–0.008)	0.001	18 §	33 §§
	Time between measurements	0.0003 (−0.00003, 0.0005)	0.076 †	17 §	
Aortic AIx@HR75 (%)	Time from COVID	0.004 (0.001–0.006)	0.003	20 §	26 §§
	Age	0.002 (−0.0001, 0.004)	0.061 †	10	

AIx, augmentation index; AP, aortic augmentation pressure; B, unstandardized regression coefficient; cDBP, central diastolic blood pressure; cfPWV, carotid–femoral pulse wave velocity; cPP, central pulse pressure; cSBP, central systolic blood pressure; P1, first systolic peak; P2, second systolic peak; PP, pulse pressure; DBP, diastolic blood pressure; MAP, mean arterial pressure; SBP, diastolic blood pressure; * Refers to the pre-COVID value of a predictor; † The predictor is not significant, or is significant at 0.1 level, but its inclusion in the multiple linear model increased adjusted R^2^ from 2 to 15%; § moderate and §§ strong association of pre–post changes with time from COVID or confounders, in accordance with the Cohen’s effect size magnitudes for R^2^ [21]; Variable is strongly correlated to the amount of time that passed between two measurements.

**Table 4 jcm-12-02123-t004:** Predictors of values of systemic and central hemodynamic parameters, as well as arterial stiffness parameters, estimated by the mixed methods regression models.

Measure:	Predictor	B (95% CI)	*p*-Value	The One-Predictor Model	The Final Model
				**Snijders/Bosker’s R^2^** **Level 1, Level 2**	
Systemic and Central Hemodynamics					
DBP (mmHg)	Time from COVID	0.20 (–0.01, 0.41)	0.063 †	8%, 9%	29% §§, 32% §§
	Age	0.32 (0.13, 0.51)	0.001	24% §, 27% §§	
cDBP (mmHg)	Time from COVID	0.19 (–0.02, 0.39)	0.082 †	7%, 8%	28% §§, 31% §§
	Age	0.33 (0.13, 0.52)	0.001	24% §, 27% §§	
MAP (mmHg)	Time from COVID	0.19 (–0.04, 0.42)	0.113 †	7%, 8%	31% §§, 34% §§
	Age	0.35 (0.17–0.53)	<0.001	27% §§, 30% §§	
Carotid–Femoral Pulse Wave Velocity					
cfPWV (m/s)	Time from COVID	0.03 (0.003, 0.05)	0.030	13% §, 15% §	28% §§, 32% §§
	Age	0.03 (0.008, 0.05)	0.005	18% §, 21% §	
	Pre–post change in cfPWV	0.19 (–0.03, 0.40)	0.094 †	1%, 0%	

B, unstandardized regression coefficient; cDBP, central diastolic blood pressure; cfPWV, carotid–femoral pulse wave velocity; DBP, diastolic blood pressure; MAP, mean arterial pressure; † The predictor is not significant or is significant at 0.1 level, but its inclusion in the mixed model increased R^2^; § moderate and §§ strong association of parameter values with time from COVID or confounders, in accordance with [21]; Level 1 defines how well the model describes changes at the level of the entire sample, whereas Level 2 depicts how well the model describes individual changes.

## Data Availability

The data presented in this study are available on request from the corresponding author A.J.

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
