# Peer review of "Long-Term Adverse Effects of Mild COVID-19 Disease on Arterial Stiffness, and Systemic and Central Hemodynamics: A Pre-Post Study"

_jcm, 2023, doi:10.3390/jcm12062123_

Round 1
Reviewer 1 Report
This is an interesting study exploring the changes in arterial stiffness, BP and other hemodynamic parameters in a small population of healthy individuals evaluated before and after COVID-19 mild infection in a pre-post study design.
The methodology for arterial stiffness and central hemodynamic measurement is, on average, appropriate. The statistical implant is accurate and well cared. The author panel included well recognized experts in the field of arterial stiffness measurement and biostatistics.
Results overall showed overall neutral changes between the two measurements in the examined parameters. The evaluation of the degree of changes led the authors to postulate a relationship with COVID-19 infection, in that they were able to demonstrate a time-dependent increase in HR-adjusted AIx changes and PWV following COVID-19 infection.
I have the following observations:
- The authors should better specify if BP was measured prior or simultaneously to arterial stiffness measurement. If this was obtained with subjects in seated or supine position and if the subject changed body posture between the two measurements. I think that the term “ambulatory” BP needs to be changed with “office” BP.
- The following list of predictors could be extrapolated from the text: age, sex, the amount of time that passed since the start of COVID infection, the amount of time that passed between the pre- and post-COVID-19 measurements, pre-COVID baseline values, and the type of device. Did the authors also check for the significance of individual HR changes between visits? Was it the same for individual BMI changes?
- How mean brachial BP was measured? Was it directly measured or estimated? If estimated, this should be included among the limitations (e.g. Grillo A et al, J Hypertens. 2020; 38:2161-2168).
- How did the authors explain that the longer the period from COVID-19 infection, the higher the PWV change? This appears, at a first sight, slightly contra-intuitive given that the inflammation burden associated with COVID-19 is expected to decrease (not to increase) over time. What is the biological plausibility of this process?
- The authors should also make a comment on the biological significance of observed COVID-19-induced changes in hemodynamic parameters. Are they clinically relevant, from their standpoint? For example, from a visual analysis of Figure 1, a reader would deduce that changes observed in 40 days following COVID-19 are of the same degree of changes observed in 40 years in a healthy individual…
- Could the authors add some information about other concomitant risk factors (e.g. smoking, metabolic syndrome or DM, dyslipidemia, familial history of CV disease) of the examined population?
Minor points
- In the discussion, the sentence “In fact, except for cfPWV at the significance level of 0.1, we were unable to …” could be misinterpreted. I suggest to change with “in fact, except for a non-significant trend for cfPWV, we were unable to…”
- I also suggest to change the following sentence “while our result support the role of age…” with “while our result suggest the role of age…”
- It seems that the process described in Figure 3 is not related to COVID-19 infection. If so, this Figure could be removed.
Reviewer 2 Report
The researchers examined whether there were individual changes in atherosclerosis before and after infection in patients diagnosed with mild COVID-19. They concluded that the vasculature may have an extensive and long-lasting pathological process following mild COVID-19 disease. Researchers should appreciate the novelty of this study. However, there are some problems with the manuscript.
Introduction
Please state the main hypothesis of the study.
Results
Please provide a flowchart showing the number of patients randomized, the number of patients who received the intended treatment, and the number of patients included in the analysis of the primary endpoint for each arm.
Other Information
Provide the registration agency and registration number of the clinical trial. If not pre-registered, the reason for not pre-registering should be clarified.
Round 2
Reviewer 1 Report
All comments have been addressed